# Dietary Supplementation with Lysozyme–Cinnamaldehyde Conjugates Enhances Feed Conversion Efficiency by Improving Intestinal Health and Modulating the Gut Microbiota in Weaned Piglets Infected with Enterotoxigenic *Escherichia coli*

**DOI:** 10.3390/ani13223497

**Published:** 2023-11-13

**Authors:** Zhezhe Tian, Jiaming Chen, Tongbin Lin, Junhua Zhu, Haoyang Gan, Fang Chen, Shihai Zhang, Wutai Guan

**Affiliations:** 1Guangdong Province Key Laboratory of Animal Nutrition Control, College of Animal Science, South China Agricultural University, Guangzhou 510642, China; tzz197211@163.com (Z.T.); cjm20221025001@stu.scau.edu.cn (J.C.); 13616905519@stu.scau.edu.cn (T.L.); zju.scau@stu.scau.edu.cn (J.Z.); ganhaoyang@stu.scau.edu.cn (H.G.); chenfang1111@scau.edu.cn (F.C.); 2College of Animal Science and National Engineering Research Center for Breeding Swine Industry, South China Agricultural University, Guangzhou 510642, China; 3Guangdong Laboratory for Lingnan Modern Agriculture, South China Agricultural University, Guangzhou 510642, China

**Keywords:** lysozyme, cinnamaldehyde, *Escherichia coli* K88, piglet, intestinal health

## Abstract

**Simple Summary:**

Intestinal health is an important indicator of the healthy growth of piglets. As the agricultural industry eliminates the use of antibiotics in animal feed, alternatives to antibiotics will be needed. This study aims to evaluate the efficacy of lysozyme–cinnamaldehyde conjugates (LC) as a potential alternative to antibiotics in treating piglets infected with enterotoxigenic *Escherichia coli* (ETEC). The result shows that the LC-supplemented diet effectively mitigated the adverse effects of *E. coli* K88, including intestinal barrier damage and inflammation. Furthermore, it improved the structure of the intestinal flora, ultimately contributing to better growth performance in piglets. LC could be a useful, safe, and natural anti-inflammatory feed additive to prevent the decline in piglets’ intestinal health.

**Abstract:**

This study aims to evaluate the efficacy of lysozyme–cinnamaldehyde conjugates (LC) as a potential alternative to antibiotics in treating piglets infected with enterotoxigenic *Escherichia coli* (ETEC). The results demonstrated that piglets fed with the LC diet exhibited lower rectal temperature and fecal scores at 9 h, 24 h, and 48 h post-ETEC challenge. Furthermore, LC supplementation led to significant improvements in the mechanical and immune barriers of the jejunum and ileum, as indicated by an increased villi-height-to-crypt-depth ratio (VCR) and the expression of tight junction proteins, mucin, and β-defensins. Furthermore, the LC diet lowered the levels of pro-inflammatory cytokines TNF-α and IL-1β in the plasma. Further analyses showed that the LC diet downregulated genes (specifically TLR4 and MyD88) linked to the TLRs/MyD88/NF-κB signaling pathway in the small intestine. Additionally, 16SrDNA sequencing data revealed that LC supplementation increased the α diversity of intestinal microorganisms and the relative abundance of *Lactobacillus*. In summary, the LC-supplemented diet effectively mitigated the adverse effects of *E. coli* K88, including intestinal barrier damage and inflammation. Furthermore, it improved the structure of the intestinal flora, ultimately contributing to better growth performance in piglets.

## 1. Introduction

Enterotoxigenic *Escherichia coli* (ETEC) is the primary cause of piglet post-weaning diarrhea (PPWD), a condition that results in high mortality rates. PPWD can result in decreased production performance, leading to high mortality rates [1]. Traditionally, antibiotics have been used in the diet to prevent PPWD and improve growth performance [2]. However, the overuse of antibiotics has led to the spread of drug-resistant bacteria [3,4]. Research has also shown the worldwide emergence of pathogens that are resistant to colistin and methicillin-resistant *Staphylococcus aureus*, posing a considerable threat to public health [5]. To address this issue, many countries have enacted laws restricting the use of antibiotics in animal feed [6]. As a result, there is an urgent need to explore alternatives to antibiotics in the form of functional feed additives.

Lysozyme, also known as N-acetyl cell wall glycan hydrolase or cell wall enzyme, is a natural anti-inflammatory, antibacterial, antiviral, and immune-enhancing agent [7]. In animal husbandry, lysozyme is a common feed additive due to its beneficial effects on growth and feed efficiency [8]. Research indicates that lysozyme supplementation significantly improves the apparent total tract digestibility of dry matter and gross energy [9], increases the villus height/crypt depth ratio [10], and reduces the serum levels of IL-1 and IL-6 in ETEC-infected weaned piglets [11]. However, the practical application of lysozyme is limited due to its instability and relatively narrow spectrum of antibacterial activity [12]. As a result, it is often combined with other antimicrobial agents to enhance its effectiveness [13].

Cinnamaldehyde, a bioactive compound found in Cinnamomum plants, is known for its distinctive aromatic odor. Pharmacological studies have demonstrated that Cinnamomum exhibits numerous beneficial activities, including antioxidant, antiviral, anti-inflammatory, antibacterial, and immunomodulatory effects [14]. Furthermore, recent research has revealed that the dietary inclusion of cinnamaldehyde effectively inhibits the NF-κB signaling pathway, thereby enhancing intestinal health in early weaned rats [15].

Lysozyme–cinnamaldehyde conjugates (LC) are formed through the reduction coupling effect of cinnamaldehyde on lysozyme. This process involves the combination of the aldehyde group in cinnamaldehyde with the amino acid residue of lysozyme, resulting in the formation of an imine bond. The conjugate has the potential to enhance the antibacterial capability of lysozyme and improve its stability. Therefore, the objective of this study was to investigate the effects of dietary LC on protecting weaning piglets challenged with ETEC, and to investigate the underlying mechanisms involved.

## 2. Materials and Methods

### 2.1. Animal Experimental Design

The animal test programs in this study strictly comply with the Protection and Use of Experimental Animals guidelines issued by the Animal Protection and Use Committee of South China Agricultural University (2022g028, Guangzhou, China). Thirty weaned piglets (Duroc × Landrace × Yorkshire, half male and half female, 24 ± 1 d, 7 ± 0.5 kg) were selected and divided into five groups based on similar body weight, with six replications in each group. The experimental groups were designed as follows: (1) basal diet (CON); (2) basic diet with ETEC (CON + ETEC); (3) basal diet with antibiotics and ETEC (50 mg/kg of quinenone, 75 mg/kg of chlortetracycline, 50 mg/kg of kitasamycin; ATB + ETEC); (4) basal diet with 0.05% LC (lysozyme/cinnamaldehyde = 1:3; Beijing Agrobeta Biotech Co., Ltd. (Beijing, China); effective content of 50,000 U/g; LC); (5) basal diet with 0.05% LC and ETEC (LC + ETEC). The basic diet formula of the experiment was prepared according to the recommendations of the National Research Council (NRC, 2012). The basal diets were mash, and the diet composition and nutrition level were shown in Table 1. Prior to the experiment, the pig house and the metabolic cage underwent thorough cleaning and disinfection using 20% concentrated glutaraldehyde. The piglets were placed in a metabolic cage, provided with free access to food and water, and the temperature was maintained at 30 ± 2 °C with a humidity level below 80%. The design of the animal experiment is shown in Appendix A.

### 2.2. Bacteria Preparation and Oral Challenge

Enterotoxigenic *Escherichia coli* (CVCC225, serotype O149: K91: K88ac; toxin LT and ST) was purchased from the China Veterinary Preservation Center. The ETEC was cultured overnight in a nutrient broth (NB, Huankai, Guangzhou, China) and subsequently diluted to a final concentration of 5 × 10^9^ CFU/mL for the challenge piglets. After a three-day adaptation period, each group of piglets was fed with corresponding diets for seven days. On the eighth day, the CON + ETEC group, ATB + ETEC group, and LC + ETEC group were fed with 10 mL (5 × 10^9^ CFU/mL) of the *E. coli* K88 solution. The CON group and LC group were fed with the same amount of liquid medium and continued to feed for seven days.

### 2.3. Fecal Score and Rectal Temperature

All piglets’ rectal temperatures and fecal scores were measured using an electronic rectal thermometer (Omron MC-347, Omron, Dalian, China) before and 9 h, 24 h, 48 h, 96 h, and 168 h after the challenge with ETEC according to the fecal scoring system reported by Marquardt et al. [16]. (0: normal, hard form; 1: soft feces, soft form; 2: mild diarrhea, no form, loose, puddles; 3: severe diarrhea, watery).

### 2.4. Production Performance

The piglets were weighed on days 0, 8, and 15, and their daily feed intake was consistently recorded throughout the trial. These measurements were used to calculate the average daily gain (ADG), average daily feed intake (ADFI), and feed conversion rate (F/G) of the piglets.

### 2.5. Sample Collection

Blood samples were collected from each piglet’s anterior vena cava on the morning of the 15th day of the trial period. A volume of 5 mL of blood was collected from each piglet using an EDTA anticoagulant tube. The collected blood samples were centrifuged at 1000× *g* for 15 min to obtain plasma samples. Simultaneously, fresh feces from all piglets were collected and stored at −80 °C in a 5 mL aseptic EP tube for testing. Well-shaped areas (2 cm) of the duodenum, jejunum, and ileum were cut off, washed with normal saline, and placed in transparent plastic bottles containing 4% paraformaldehyde at room temperature. Additionally, 2 cm tissue samples from the duodenum, jejunum, and ileum were carefully washed with normal saline. The mucous membrane was then scraped off with a glass plate and placed in a 5 mL cryopreservation tube. The samples were quickly frozen using liquid nitrogen and stored at −80 °C. It is important to note that all samples were taken from the same part of each piglet.

### 2.6. Measurement of Inflammatory Factor

The concentrations of IL-1β (CSB-E06782p, Jiancheng, Nanjing, China), TNF-α (CSB-E16980p, Jiancheng, Nanjing, China), IL-6 (CSB-E06786p, Jiancheng, Nanjing, China), and IL-10 (CSB-E06779p, Jiancheng, Nanjing, China) in plasma were measured using an ELISA kit. The measurements were conducted following the instructions provided by the kit manufacturer.

### 2.7. Intestinal Section

The fixed intestinal tissue samples were extracted from a 4% paraformaldehyde solution. They were then graded, dehydrated in a specific order, embedded, and sliced to a thickness of 5 μm. The slices were stained with hematoxylin–eosin staining solution, dehydrated, and sealed. For each sample, two slices were prepared, and three typical visual fields (40 times magnification) were selected. The height of the intestinal villi and the depth of the crypt were determined using image processing software.

### 2.8. Real-Time Fluorescence Quantitative PCR Detection

RNA extraction from the intestinal tissue was performed using the tissue RNA purification kit (EZB-RN001-plus, EZBioscience, Roseville, MN, USA). The concentration and quality of the extracted RNA were determined using a NanoDrop ND-1000 spectrophotometer (Thermo Scientific, Wilmington, NC, USA). The RNA was then reverse transcribed into cDNA using a reverse transcriptase kit (A0010CGQ, EZBioscience, Roseville, MN, USA). Finally, the expression of the target gene was analyzed using the 2 × Color SYBR Green qPCR Master Mix (ROX2 plus) kit (A0012-R2, EZBioscience, Roseville, MN, USA). Real-time quantitative PCR detection was carried out using the 7500 fast real-time PCR system with a reaction condition of 95 °C for 5 min. The qPCR reaction system consisted of a 20 μL volume with 9.2 μL of the cDNA and ddH2O premixture, 2× Color SYBR Green qPCR Master Mix, and primer pair premixture, and 10.8 μL for each reaction well. The amplification was performed for 40 cycles at 95 °C for 10 s, 60 °C for 30 s, and 72 °C for 30 s. The primer sequences we used for the real-time PCR are listed in Table 2.

### 2.9. Western Blotting Analysis

The level of protein expression of the tight junction proteins (Claudin-1, Occludin, and ZO-1) was measured using Western blotting. The total protein was isolated using RIPA lysate buffer (Beyotime, Shanghai, China). A total of 10 μg of protein was separated using a 10% SDS-PAGE (P0012AC, Beyotime, Shanghai, China) and transferred to a PVDF membrane for Western blotting. The membrane was then incubated with a specific antibody for 12 h at 4 °C, followed by incubation with a secondary antibody for 1 h at room temperature. Protein bands were detected using an ECL chemiluminescence reagent (P1020 ApplyGen, Beijing, China). The test antibody information is shown in Table 3.

### 2.10. 16S rDNA Sequencing

The intestinal microflora of the piglet feces was analyzed using a high-throughput 16s rDNA sequencing technique. The sequencing was conducted by Novogene Denovo in Guangzhou, China. Genomic DNA was extracted from the samples, and the V3 + V4 region of 16s rDNA was amplified using specific primers with a barcode. The primer sequence used was 341F: CCTACGGGNGGCWGCAGTATCTAAT. The purified amplified products were then connected to the sequencing connector, and the sequencing library was constructed and sequenced using Illumina. Finally, the 16s rDNA sequencing data were analyzed using Omicsmart, an online tool provided by Gene Denovo in Guangzhou, China.

### 2.11. Statistical Analysis

By using the SPSS 26.0 software (IBM Inc., Armonk, NY, USA) to analyze the one-way ANOVA variance of all data and using Duncan’s method to make multiple comparisons, the results were expressed as means ± SEM, and the statistical significance standard was *p* < 0.05, It was considered as a trend when 0.05 < *p* < 0.10. Graphs were drawn and analyzed using the GraphPad Prism 8.0 software.

## 3. Results

### 3.1. Production Performance, Fecal Scores, and Rectal Temperature

As illustrated in Table 4, the CON + ETEC group showed a significant decrease in average daily gain (ADG) and an increase in the feed-to-gain ratio (F/G) when compared to the CON group (*p* < 0.05). No significant difference was observed among the other groups. As depicted in Figure 1A, the fecal score of the CON + ETEC group rose significantly at 9 h, 24 h, 48 h, and 96 h post-challenge in comparison to the CON group (*p* < 0.05). At 24 h post-challenge, piglets in the ATB + ETEC, LC + ETEC, and LC treatments had lower fecal scores compared with those from the CON + ETEC (*p* < 0.05) (Figure 1A). Furthermore, Figure 1B indicates that the rectal temperature of the CON + ETEC group significantly increased at 9 h, 24 h, and 48 h post-challenge relative to the CON group (*p* < 0.05). Compared with the CON + ETEC group, the rectal temperature of the ATB + ETEC group and LC + ETEC group decreased significantly at 24 h after the challenge (*p* < 0.05) (Figure 1B). And there was no significant change between the LC + ETEC group and the ATB + ETEC group.

### 3.2. Inflammatory Cytokines

As depicted in Figure 2, compared to the CON group, the plasma levels of TNF-α, IL-1β, and IL-6 in the CON + ETEC group increased significantly post-challenge (*p* < 0.05). Conversely, no significant changes were observed in the ATB + ETEC group or LC + ETEC group. Importantly, Figure 2D indicates that the TNF-α levels in the LC group decreased significantly relative to other groups (*p* < 0.05).

### 3.3. Gut Morphology

Intestinal morphology was examined using H&E staining in the weaned piglets (Figure 3A). Compared to the CON group, the depth of the duodenal recess increased in the CON + ETEC group, while the villus height of the jejunum and ileum, as well as the VCR of each intestinal segment, decreased significantly (*p* < 0.05) (Figure 3B–D). Additionally, the duodenal VCR increased significantly in the ATB + ETEC group compared to the CON + ETEC group (*p* < 0.05) (Figure 3D). And the duodenal VCR of the LC + ETEC group showed an upward trend compared to the CON + ETEC group (*p* = 0.061) (Figure 3D). Furthermore, the jejunum and ileum VCR increased significantly in both the ATB + ETEC group and the LC + ETEC group (*p* < 0.05) (Figure 3D).

### 3.4. Intestinal Tight Junctions

Similarly, the CON + ETEC group showed a significant decrease in tight junctions across all intestinal segments compared to the CON group (*p* < 0.05) (Figure 4A–F). Notably, the supplementation of LC into the diet led to an increase in Occludin expression throughout the intestinal tract relative to the CON treatment (*p* < 0.05) (Figure 4A–F). In addition, the protein expressions of Occludin, Claudin-1, and ZO-1 in the duodenum, Occludin and Claudin-1 in the jejunum, and Occludin in the ileum were significantly higher in the ATB + ETEC and LC + ETEC groups compared to the CON + ETEC group (*p* < 0.05) (Figure 4A–F). And there were no significant differences in intestinal tight junction protein expression between the ATB + ETEC and the LC + ETEC groups. Full original Western blot (WB) images of Figure 4 are shown in Appendix A.

### 3.5. Mucin and β-Defensin

Compared to the CON group, the CON + ETEC group exhibited a significant reduction in the expression levels of *pBD2* in the duodenum, as well as *MUC-1*, *MUC-2*, *pBD1*, and *pBD2* in the jejunum, and *MUC-2*, *pBD1*, *pBD2*, and *pBD3* mRNA in the ileum (*p* < 0.05) (Figure 5A–F). In contrast, the ATB + ETEC and LC + ETEC groups showed no significant down-regulation in the expression of mucin and β-defensin genes (Figure 5A–F). Furthermore, the LC group displayed a marked increase in the mRNA expression levels of *pBD2* in the jejunum, as well as *MUC-2*, *pBD1*, and *pBD2* in the ileum, when compared to the CON group (*p* < 0.05) (Figure 5E,F).

### 3.6. Abundance of Genes Related to NF-κB/MAPK Signal Pathway

As illustrated in Figure 6, the CON + ETEC group showed a significant up-regulation in the mRNA expression of TLR4 and MyD88 in the duodenum, jejunum, and ileum when compared with the CON group (*p* < 0.05) (Figure 6A–C). Furthermore, the mRNA expression of IKKα and TAK1 in the duodenum, TAK1 in the jejunum, and TRAF6 in the ileum were also significantly up-regulated in the CON + ETEC group (*p* < 0.05) (Figure 6A–C). In contrast, the ATB + ETEC and LC + ETEC groups did not display a significant increase in the expression of NF-κB pathway-related genes. However, the mRNA expression of SIGTRR was significantly up-regulated (*p* < 0.05) in both of these groups (Figure 6A–C). Simultaneously, the LC group showed a significant down-regulation in the mRNA levels of TLR4, IκB, and TRAF6 in the duodenum, as well as TLR4 and TAK1 in the jejunum, and MyD88, TAK1, and TRAF6 in the ileum (*p* < 0.05) (Figure 6A–C). No significant difference in gene expression was observed between the ATB + ETEC and LC + ETEC groups.

### 3.7. Gut Microbiome

A total of 3,672,934 high-quality sequences were generated from 30 fecal samples (five treatments; *n* = 6); with an average of 122,431 sequences per sample. After removing the noise sequence, 2,956,668 valid tags remained for further analysis. These tags were clustered into operational taxons (OTUs) with 97% sequence similarity. These OTUs were then classified to 22 phyla; 29 classes; 47 orders; 68 families; 170 genera; and 1395 OTUs (97% similarity) for further analysis. The alpha diversity of the ileal microbiota, as indicated by Sob, Chao1, ACE, Shannon, Simpson, and Pielou, is presented in Figure 7A–F. Notably, the CON + ETEC treatment group showed a significant reduction in Chao1 and ACE indices compared to the other groups (*p* < 0.05). Furthermore, no significant differences were found in the beta diversity of ileal microbiota among the treatment groups (Figure 7G,H). The microbial composition of the piglets’ ileal contents predominantly consisted of five major bacterial phyla—Firmicutes; Bacteroidetes; Proteobacteria; Cyanobacteria; and Epsilonbacteraeota—which collectively made up over 95% of the total ileal bacterial community (Figure 8A). However; the proportions of these phyla varied among the different treatments. At the genus level, the relative abundance of *Lactobacillus* in piglets was observed to decrease following the ETEC challenge. Conversely, piglets supplemented with LC and ATB exhibited an increase in *Lactobacillus* abundance (*p* < 0.05) (Figure 8B).

## 4. Discussion

ETEC is a major contributor to diarrhea and intestinal injury in weaned piglets, as mentioned in previous studies [17]. Specifically, the heat instability (LT) and heat-stable (ST) enterotoxins produced by ETEC increase the levels of cyclic adenosine monophosphate (cAMP) and cyclic guanosine monophosphate (cGMP). This leads to an increased secretion of water and electrolytes into the intestinal cavity, culminating in diarrhea in piglets [18]. In this study, we observed a significant rise in both rectal temperature and fecal score in piglets 9 h post-ETEC challenge, followed by a gradual decline. These observations highlight the successful establishment of the ETEC challenge model.

The present study demonstrates that piglets in the dietary LC group displayed lower rectal temperatures and fecal scores, mimicking the beneficial effects observed in the ATB group. These results suggest that LC effectively mitigates ETEC-induced diarrhea in piglets. Previous studies have shown that cinnamaldehyde and lysozyme are known to exert potent bactericidal effects against both Gram-negative and Gram-positive bacteria [19]. Additionally, lysozyme–ZnO nanoparticle coupling [20] and cinnamaldehyde [21] have been found to possess significant antibacterial properties against *E. coli*. These findings strongly suggest that LC has the capacity to inhibit *E. coli* colonization in the intestinal tract. Furthermore, it was observed that the feed-to-gain ratio (F/G) of piglets significantly increased after the challenge, but LC was able to effectively alleviate this result. There is growing evidence that the improvement in the F/G in piglets by LC is closely related to their intestinal health. Garas observed that feeding goat milk rich in lysozyme to pigs infected with ETEC F4 can reduce the incidence of diarrhea and improve their production performance [22].

To investigate the impact of LC on the intestinal health of piglets, we initially assessed the intestinal morphology across all treatment groups. The investigation revealed that different intestinal segments suffered varying degrees of damage after the ETEC challenge, with the jejunum and ileum exhibiting more severe damage. This could be attributed to the fact that ETEC primarily colonizes the posterior intestinal segment [23]. Both LC and ATB were found to effectively alleviate the damage caused by ETEC. As previous research mentioned, lysozyme has been identified as a viable alternative to antibiotics for improving small intestine morphology in pigs, which aligns with the results of our study. Notably, our results suggest an even greater protective effect provided by lysozyme when modified with cinnamaldehyde. It is important to note that intestinal morphology serves as the foundation for intestinal structure and function. Any damage to it not only leads to reduced nutrient absorption in piglets but also serves as an indicator of inflammation and impairment of the intestinal barrier [24].

In this study, we observed a marked down-regulation in the protein expression levels of Occludin, Claudin-1, and ZO-1, as well as in the mRNA expression levels of mucin and defensin genes, in various intestinal segments of ETEC-challenged piglets. Importantly, supplementation with LC mitigated these adverse effects. Several studies have also demonstrated the beneficial effects of lysozyme and cinnamaldehyde in improving the function of the intestinal physical barrier in weaned piglets. For example, Huang et al. reported that human lysozyme positively influenced intestinal morphology and preserved the integrity of tight junctions, especially in the jejunum and ileum [13]. Similarly, Mo demonstrated that daily supplementation with microencapsulated compounds of thymol, carvol, and cinnamaldehyde improved tight junction protein mRNA expression and thereby enhanced intestinal barrier function in weaned piglets [25]. Consistent with these studies, our results also show that LC significantly increased the mRNA expression of *pBD-2* in the duodenum, jejunum, and ileum of piglets. *pBD-2* possesses a broad antibacterial spectrum that is effective against *E. coli*, *Salmonella typhimurium*, and *Staphylococcus aureus*. Additionally, the expression of *pBD-2* contributes to elevating the levels of intestinal tight junction proteins and mucin genes or proteins [26].

In order to further clarify the molecular mechanisms underlying LC’s mitigating effects on intestinal inflammation, we examined the TLRs/MyD88/NF-κB signal pathway [27], a classical inflammatory cascade. We also assessed its negative regulatory factors, specifically the Toll-interacting protein and Single Ig IL-1-related receptor (TOLLIP and SIGIRR) [28]. The activation of the TLRs pathway is known to be linked to multi-layered inflammatory intestinal diseases [29]. Our study observed a significant increase in the levels of IL-1β and TNF-α in plasma, as well as the expression of certain genes related to the TLRs/MyD88/NF-κB inflammatory pathway, and the mRNA expression of SIGIRR in the small intestine after the challenge. However, in piglets supplemented with either ATB or LC, these inflammatory pathway genes remained largely unaltered. These observations indicate LC’s potential in attenuating intestinal inflammation.

It is widely recognized that the intestinal tracts of animals contain a high density and diverse range of microorganisms [30]. These microorganisms play a crucial role in various aspects such as intestinal development, immune defense, and the maintenance of the intestinal mucosal barrier. These microbial communities substantially shape the production performance and intestinal well-being of weaned piglets [31]. In this study, the analysis of 16S rDNA sequencing revealed a significant decrease in fecal microbial alpha diversity in piglets infected with ETEC. In contrast, no such shift was detected in the microbial diversity of piglets subjected to diets enriched with LC and ATB. This suggests that dietary LC potentially mirrors the microbial diversity-protective effect of antibiotics. Crucially, we identified a decrease in the relative abundance of *Lactobacillus* in piglets after the ETEC challenge, but this increased in those supplemented with LC and ATB. *Lactobacillus*, which constitutes the largest number of beneficial bacteria in the intestine, plays a crucial role in fermenting carbohydrates to produce lactic acid. This helps to reduce the intestinal pH and effectively prevent the reproduction of harmful bacteria [32]. Previous research has also shown that *Lactobacillus* can fine-tune the composition of gut flora [33], boost the expression of intestinal tight junction proteins and mucins [34,35], modulate the intestinal immune responses [36], and trigger the expression of *pBD* [37]. These findings Fprovide significant evidence that LC not only decreases intestinal inflammation but also fortifies the intestinal barrier function.

## 5. Conclusions

Dietary LC could alleviate the inflammatory reaction and intestinal villus damage caused by *E. coli* K88 in weaned piglets. These benefits are conferred through multiple mechanisms, including the inhibition of the NF-κB signaling pathway, the upregulation of β-defensin gene expression, and the enhancement of the structure and growth performance of the intestinal microbiota. These findings strongly suggest that LC could enhance the growth performance and intestinal health of weaned piglets and serve as an effective alternative to antibiotics in the future.

## Figures and Tables

**Figure 1 animals-13-03497-f001:**
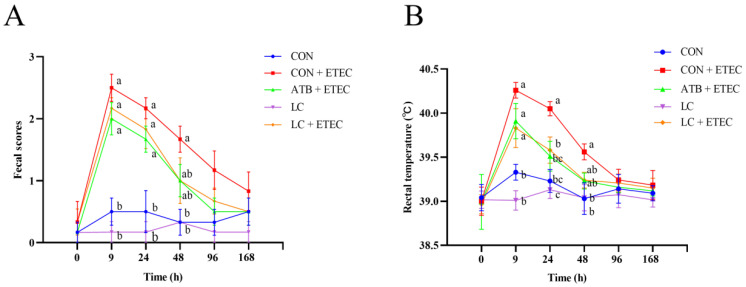
Effects of dietary LC on the fecal scores and rectal temperature in ETEC-challenged piglets. (**A**) The fecal scores were evaluated at 0 h, 9 h, 24 h, 48 h, 96 h, and 168 h after challenge with ETEC according to the fecal scoring system reported by Marquardt et al. (0: normal, hard form; 1: soft feces, soft form; 2: mild diarrhea, no form, loose, puddles; 3: severe diarrhea, watery). (**B**) Rectal temperatures were measured using an electronic rectal thermometer (Omron MC-347) at 0 h, 9 h, 24 h, 48 h, 96 h, and 168 h in ETEC-challenged piglets. Data are expressed as the mean ± SE (*n* = 6). Different letters (a, b, and c) indicate a significant difference among the different groups (*p* < 0.05).

**Figure 2 animals-13-03497-f002:**
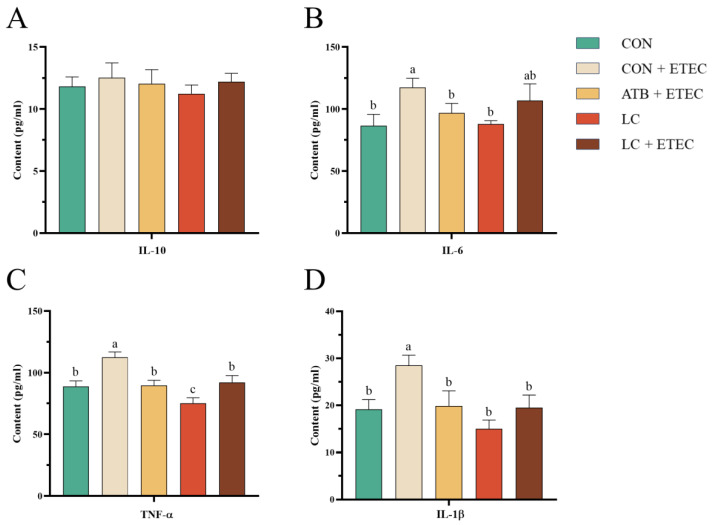
Effects of dietary LC on plasma inflammatory factors in ETEC-challenged piglets. Concentrations of IL-10 (**A**), IL-6 (**B**), TNF-α (**C**), and IL-1β (**D**) were assessed. Data are expressed as the mean ± SE (*n* = 6). Different letters (a, b, and c) indicate a significant difference among the different groups (*p* < 0.05).

**Figure 3 animals-13-03497-f003:**
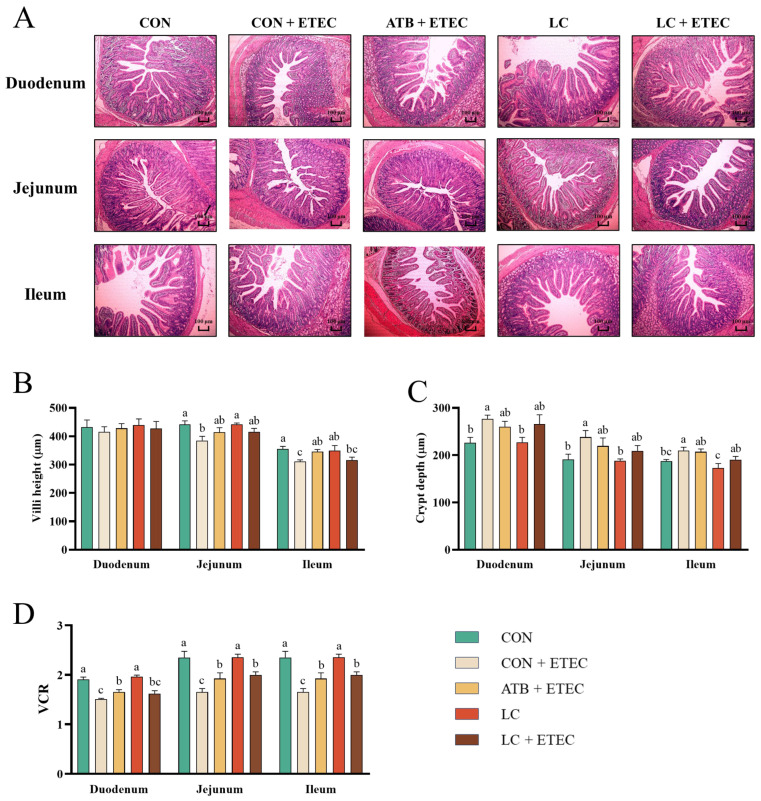
Effects of dietary LC supplementation on the intestinal morphology in ETEC-challenged piglets. (**A**) The hematoxylin and eosin (H&E)-stained intestinal cross-sections in piglets are shown. The scale bar represents 100 μm. (**B**) Villi height, (**C**) crypt depth, (**D**) ratio of villous height to crypt depth (VCR). Data are expressed as the mean ± SE (*n* = 6). Different letters (a, b, and c) indicate a significant difference among the different groups (*p* < 0.05).

**Figure 4 animals-13-03497-f004:**
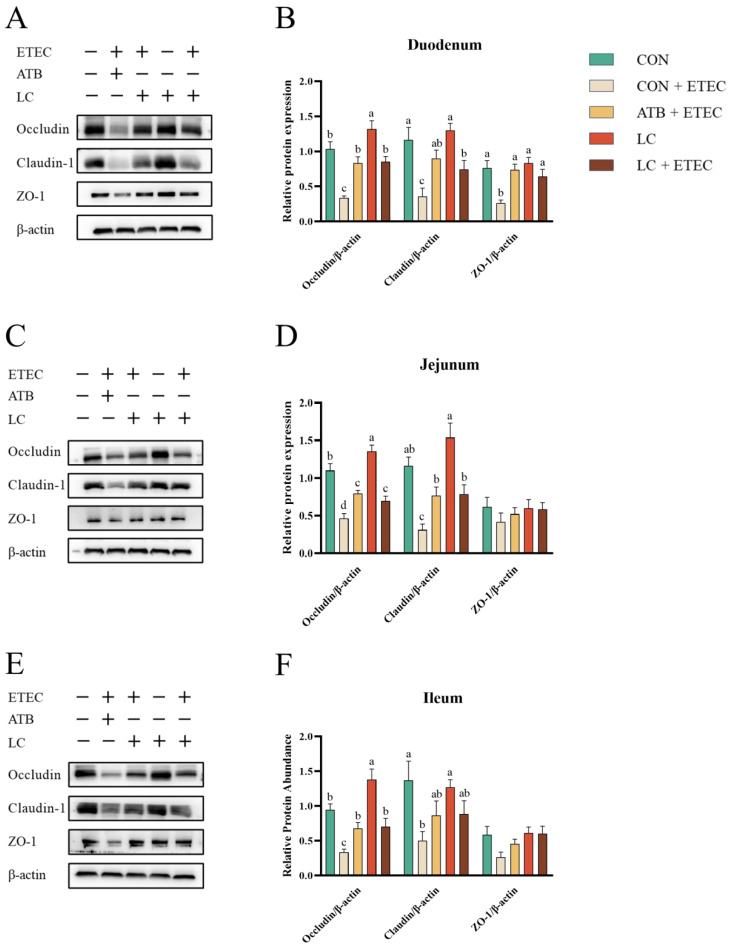
Effects of dietary LC supplementation on the expression of intestinal tight junction proteins in ETEC-challenged piglets. The expression and abundance of tight junction proteins (Occludin, Claudin-1, and ZO-1) and β-actin in the duodenum (**A**,**B**), jejunum (**C**,**D**), and ileum (**E**,**F**) of piglets were analyzed. Data are expressed as the mean ± SE (*n* = 3). Different letters (a, b, and c) indicate a significant difference among the different groups (*p* < 0.05).

**Figure 5 animals-13-03497-f005:**
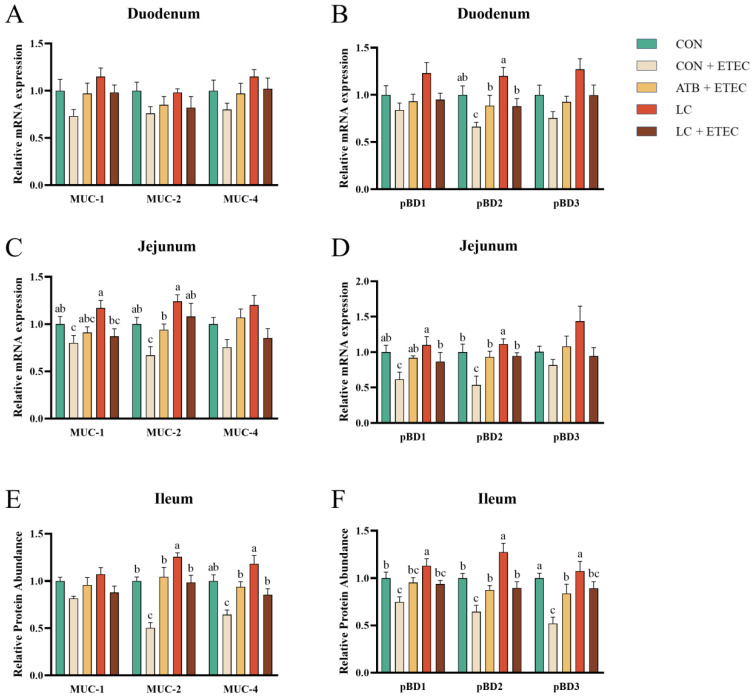
Effects of dietary LC supplementation on mRNA expression levels of mucin and β-defensin of piglets challenged with ETEC. The mRNA expression levels of mucin in the duodenum are presented in (**A**), while the expression levels of β-defensin are shown in (**B**). Similarly, (**C**) represents the expression of mucin mRNA in the jejunum, and (**D**) represents β-defensin expression. Additionally, (**E**,**F**) display the expression levels of mucin and β-defensin mRNA in the ileum. The data is presented as the mean ± SE (*n* = 6). Different letters (a, b, and c) indicate a significant difference among the different groups (*p* < 0.05). MUC, mucin; *pBD*, porcine beta-defensin.

**Figure 6 animals-13-03497-f006:**
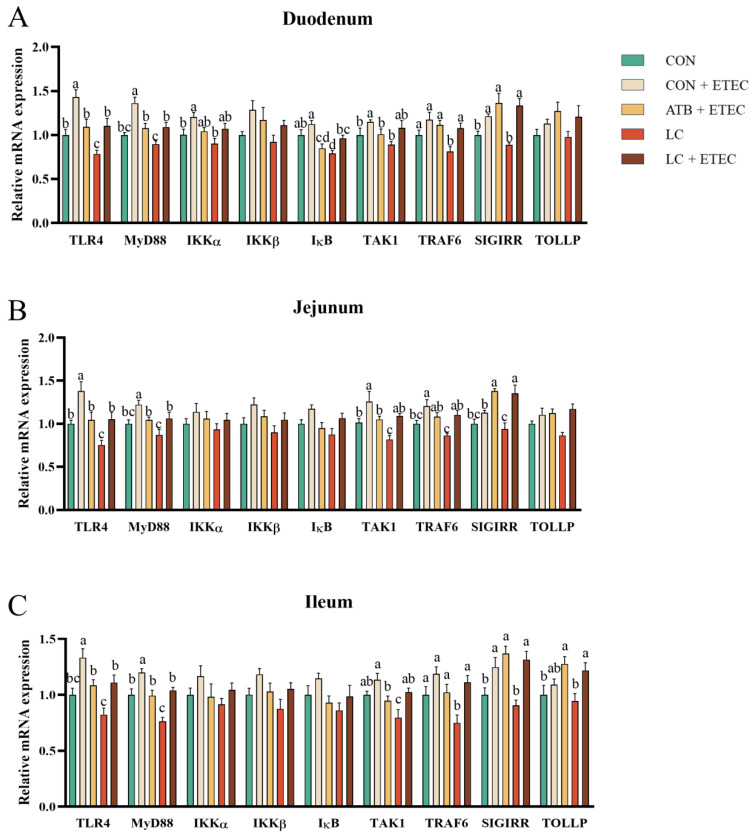
Effects of dietary LC supplementation on the concentration of relative mRNA expression of genes related to inflammation in ETEC-challenged piglets. (**A**) Duodenum, (**B**) jejunum, (**C**) ileum. The data are presented as the mean ± SE (*n* = 6). Different letters (a, b, and c) indicate a significant difference among the different groups (*p* < 0.05). TLR4, Toll-like receptor 4; NF-κB, nuclear factor kappa B; MyD88, myeloid differentiation factor 88; IKKα, inhibitor of NF-κB kinase α; IKKβ, inhibitor of NF-κB kinase β; TAK1, transforming growth factor β (TGF-β)-activated kinase 1; TRAF6, TNF receptor-associated factor 6; SIGTRR, Single Ig IL-1-related receptor; TOLLP, Toll-interacting protein.

**Figure 7 animals-13-03497-f007:**
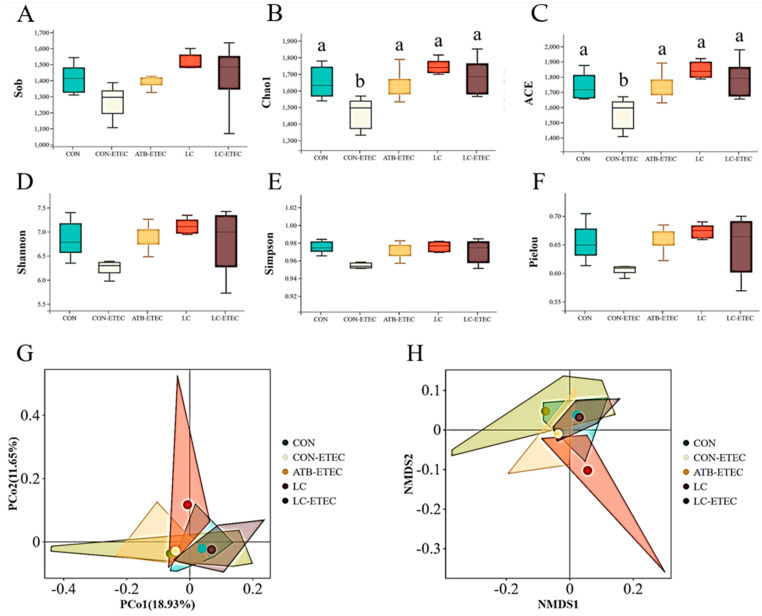
Effects of dietary on alpha diversity and beta diversity of ileal microbiota in ETEC−challenged piglets. (**A**–**F**) Alpha diversity index: Sob, Chao1, ACE, Shannon, Simpson, and Pielou. (**G**) Principal coordinates analysis based on Bray−Curtis distance. (**H**) Plot of NMDS analysis of Bray. Data are expressed as the mean ± SE (*n* = 6). Different letters (a, and b indicate a significant difference among the different groups (*p* < 0.05).

**Figure 8 animals-13-03497-f008:**
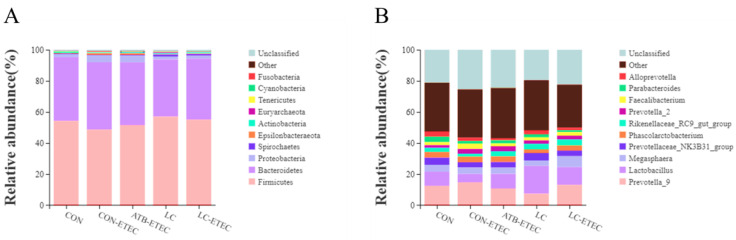
Effects of dietary LC supplementation on the microbial composition of ileal microbiota in ETEC-challenged Piglets. (**A**) Relative abundance in genus. (**B**) Relative abundance in species. Data are expressed as the mean ± SE (*n* = 6).

**Table 1 animals-13-03497-t001:** Compositions of basal diet and nutrient profile.

Ingredient Composition	Content (%)	Calculated Nutrient Profile ^b^	Content
corn (7.4%, CP)	22.16	Digestible energy (MJ/kg)	14.83
broken rice (7.6%, CP)	31.09	Crude protein (%) ^c^	17.98
soybean meal (43%, CP)	16.0	Crude fat (%) ^c^	5.18
extruded soybean (35%, CP)	3.00	Dry matter (%) ^c^	89.51
fermented soybean (50%, CP)	5.00	Crude fiber (%) ^c^	2.17
Peruvian fish meal (65%, CP)	3.50	Ash (%) ^c^	5.55
whey (3%, CP)	7.50	Calcium (%) ^c^	0.80
glucose	1.25	Phosphorus (%) ^c^	0.68
sucrose	3.50	Available phosphorus (%) ^c^	0.48
soybean oil	2.60	Lysine (%) ^c^	1.39
calcium hydrophosphate	1.20	Digestible lysine (%) ^c^	1.25
acidifier	0.20	Digestible methionine + Cystine (%) ^c^	0.69
limestone	0.60	Digestible threonine (%) ^c^	0.74
salt	0.30	Digestible tryptophan (%) ^c^	0.21
zinc oxide (78%, Zn)	0.20		
choline chloride (50%)	0.10		
L-lysine HCl (98.5%)	0.25		
DL-methionine (99%)	0.28		
L-threonine (98.5%)	0.22		
L-tryptophan (98.5%)	0.05		
premix ^a^	1.00		

^a^ Supplied per kilogram of diet: 12,000 IU of vitamin A, 3500 IU of vitamin D_3_, 40 IU of vitamin E, 3.5 mg of vitamin K3, 18 mg of vitamin B, 10 mg of vitamin B2, 12.8 mg of vitamin B6, 0.034 mg of vitamin B12, 60 mg of nicotinic acid, 25 mg of D-pantothenic acid, 2.0 mg of folic acid, 0.26 mg of D-biotin, 180 mg of iron, 75.0 mg of copper, 1500 mg of zinc, 50.0 mg of manganese, 0.64 mg of iodine, and 0.3 mg of Selenium. ^b^ Calculated nutrient profiles were the calculated values. ^c^ Analyzed values.

**Table 2 animals-13-03497-t002:** Primers used for q-PCR analysis.

Genes	Accession	Direction	Sequences (5′–3′)
*TLR4*	NM_001113039.1	Forward	GCCATCGCTGCTAACATCATC
		Reverse	CTCATACTCAAAGATACACCATCGG
*MyD88*	NM_001099923.1	Forward Reverse	TGGTAGTGGTTGTCTCTGATGA TGGAGAGAGGCTGAGTGCAA
*IKKα*	NM_001114279.1	Forward Reverse	TCTTGATCCTCGGAAACCAG TGCTTCGGCCCATACTTTAC
*IKKβ*	NM_001099935.1	Forward Reverse	CCTCACCTTGCTGAGTGACA TCCCCACAAAGGAGGTACAG
*IκB*	EU399817.1	Forward Reverse	CTGCTCGGCAATAACACTGA GAGAGGAGACCGTTGGTGAG
*TRAF-6*	XM_013990069.2	Forward Reverse	GCTGCATCTATGGCATTTGAAG CCACAGATAACATTTGCCAAAGG
*TAK1*	NM_001114280.1	Forward Reverse	CATGTGGGCTGTTCATAACG GAGTTGCTCTGGCCTTCATC
*SIGIRR*	*NM_001315689.1*	Forward Reverse	ACCTGGGCTCCCGAAACTAC GTCATCTTCTGACACCAGGCAAT
*TOLLIP*	NM_001315800.1	Forward Reverse	CCCGCGCTGGAATAAGG CATCAAAGATCTCCAGGTAGAAGGA
*PBD-1*	NM_213838.1	Forward Reverse	ACCGCCTCCTCCTTGTATTC CACAGGTGCCGATCTGTTTC
*pBD-2*	NM_214442.2	Forward Reverse	ATTAACCTGCTTACGGGTCTTGGC CCACTGTAACAGGTCCCTTCAATCC
*pBD-3*	XM_021074698.1	Forward Reverse	TCTTCTTGTTCCTGATGCCTCTTCC GCCACTCACAGAACAGCTACCTATC
*MUC1*	XM_021089730.1	Forward Reverse	TTCTTCGGGCTGTTGCTACT ACTGTCTTGGAAGGCCAGAA
*MUC2*	NM_010843.1	Forward Reverse	CTCCATCCGCTTCAGAAAAG ACTACCTGGGGCCTCTGAAT
*MUC4*	XM_021068274.1	Forward Reverse	CCTCCCAAGCAGATGTCAAT CTGGGATAAGAATGCCTCCA
*β-Actin*	XM_021086047.1	Forward Reverse	GATCTGGCACCACACCTTCTACAAC TCATCTTCTCACGGTTGGCTTTGG

*TLR4*, Toll-like receptor 4; *MyD88*, myeloid differentiation factor 88; *IKKα*, inhibitor of NF-κB kinase α; *IKKβ*, inhibitor of NF-κB kinase β; *IκB*, Inhibitor of κ light chain gene; *TRAF-6*, TNF receptor associated factor 6; *TAK1*, transforming growth factor β activated kinase 1; *SIGIRR*, Single Ig IL-1-related receptor; *TOLLIP*, Toll-interacting protein; *pBD*, porcine beta-defensin; and *MUC*, mucin.

**Table 3 animals-13-03497-t003:** Antibodies used for Western blotting analysis.

Names	Company	Cat. No.	Dilution Multiple
Claudin-1	Abcam (Cambridge, UK)	ab15098	1:1000
Occludin	Abcam (Cambridge, UK)	ab31721	1:1000
ZO-1	Abcam (Cambridge, UK)	ab31721	1:1000
β-actin	Bioss (Beijing, China)	bs-0061R	1:5000
Second antibody (Goat anti-rabbit)	ZenBio (Chengdu, China)	511203	1:5000

**Table 4 animals-13-03497-t004:** Effect of LC on the growth performance of weaned piglets.

Items	Treatments					SEM	*p*-Value
	CON	CON + ETEC	ATB + ETEC	LC	LC + ETEC		
Average body weight (kg)							
0 d	7.10	7.09	7.08	7.06	7.11	0.06	1.00
7 d	8.75	8.71	8.81	8.80	8.85	0.05	0.91
14 d	10.97	10.58	10.85	11.05	10.86	0.10	0.40
Pre-challenge (0–7 d)							
ADFI (g)	322.27	318.19	339.28	339.17	341.74	7.48	0.81
ADG (g)	235.24	231.43	246.43	248.57	248.33	5.42	0.80
F/G	1.37	1.38	1.38	1.36	1.38	0.03	0.44
Post-challenge (7–14 d)							
ADFI (g)	481.97	460.28	452.41	486.27	444.69	11.24	0.74
ADG (g)	319.24	267.03	290.95	321.91	284.12	7.75	0.06
F/G	1.51 ^b^	1.72 ^a^	1.55 ^b^	1.50 ^b^	1.57 ^b^	0.02	0.00

*n* = 6. Different letters mean a statistically significant difference among the groups (*p* < 0.05).

## Data Availability

Data are contained within the article and Appendix A; The datasets generated for this study are available on request from the corresponding author.

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
