# Peer review of "Dietary Supplementation with Lysozyme–Cinnamaldehyde Conjugates Enhances Feed Conversion Efficiency by Improving Intestinal Health and Modulating the Gut Microbiota in Weaned Piglets Infected with Enterotoxigenic Escherichia coli"

_animals, 2023, doi:10.3390/ani13223497_

Round 1

Reviewer 1 Report

Comments and Suggestions for Authors

The study entitled "Dietary supplementation with lysozyme-cinnamaldehyde conjugates enhances growth performance by improving intestinal health and modulating the gut microbiota in weaned piglets" was designed to evaluate lysozyme-cinnamaldehyde conjugates as potential substitute for antibiotic replacement. It would have been great if the authors had included 1 antibiotic the lysozyme-cinnamaldehyde conjugates can replace. 

Additional, evaluating pig's performance with only 6 pigs per treatment does not present a very good picture and hence performance result not reliable.

Overall, the authors authors did a good job but the following needs to be fixed.

Lines 41 - 42: Delete the sentence "Mechanistically, PPWD negatively impacts intestinal villi, leads to intestinal inflammation, and impairs immune function, ultimately resulting in death" This information is provided in the preceding sentence.

Lines 44 - 46: Delete this statement. "However, the overuse of antibiotics has led to residues and the emergence of drug resistance in livestock, posing significant health risks to humans" This statement is over generalization. Could you provide a single study that was conducted to proof this statement?

Line 77: Replace the word "repeats" with "replications"

Line 83: Replace the word "basic" with "basal"

Line 83: Replace the word "powder" with "mash"

Lines 73 - 95. What is the experimental design?

Comments on the Quality of English Language

The study entitled "Dietary supplementation with lysozyme-cinnamaldehyde conjugates enhances growth performance by improving intestinal health and modulating the gut microbiota in weaned piglets" was designed to evaluate lysozyme-cinnamaldehyde conjugates as potential substitute for antibiotic replacement. It would have been great if the authors had included 1 antibiotic the lysozyme-cinnamaldehyde conjugates can replace. 

Additional, evaluating pig's performance with only 6 pigs per treatment does not present a very good picture and hence performance result not reliable.

Overall, the authors authors did a good job but the following needs to be fixed.

Lines 41 - 42: Delete the sentence "Mechanistically, PPWD negatively impacts intestinal villi, leads to intestinal inflammation, and impairs immune function, ultimately resulting in death" This information is provided in the preceding sentence.

Lines 44 - 46: Delete this statement. "However, the overuse of antibiotics has led to residues and the emergence of drug resistance in livestock, posing significant health risks to humans" This statement is over generalization. Could you provide a single study that was conducted to proof this statement?

Line 77: Replace the word "repeats" with "replications"

Line 83: Replace the word "basic" with "basal"

Line 83: Replace the word "powder" with "mash"

Lines 73 - 95. What is the experimental design?

Author Response

Dear Editor-in-chief,

Thank you very much for your and the reviewers’ effort on our manuscript (animals-2656122, entitled " Dietary supplementation with lysozyme-cinnamaldehyde conjugates enhances growth performance by improving intestinal health and modulating the gut microbiota in weaned piglets "). We have addressed the comments of the reviewer point-by-point and hope the revised version is now acceptable for publication in "Animals". The revised manuscript was marked with red color.

For your information, we have revised the manuscript accordingly as follows:

REVIEWER REPORT(S):
Referee: 1

Comments to the Author
The study entitled "Dietary supplementation with lysozyme-cinnamaldehyde conjugates enhances growth performance by improving intestinal health and modulating the gut microbiota in weaned piglets" was designed to evaluate lysozyme-cinnamaldehyde conjugates as potential substitute for antibiotic replacement. It would have been great if the authors had included 1 antibiotic the lysozyme-cinnamaldehyde conjugates can replace.

Additional, evaluating pig's performance with only 6 pigs per treatment does not present a very good picture and hence performance result not reliable.

AnswerAs the reviewer mentioned, the sample size in this experiment was limited. In the future, we will conduct animal experiments with a larger sample size in scaled-up pig farms. Additionally, we will include a positive control group using antibiotics that can serve as an alternative to the lysozyme-cinnamaldehyde conjugates for comparison.

Major revision
1. Lines 41 - 42: Delete the sentence "Mechanistically, PPWD negatively impacts intestinal villi, leads to intestinal inflammation, and impairs immune function, ultimately resulting in death" This information is provided in the preceding sentence.

Response: Thank you for your suggestions. We have deleted these sentences in line 56.

  1. Lines 44 - 46: Delete this statement. "However, the overuse of antibiotics has led to residues and the emergence of drug resistance in livestock, posing significant health risks to humans" This statement is over generalization. Could you provide a single study that was conducted to proof this statement?

Response: Thank you for your suggestions. We have deleted these sentences. According to this survey,  we have provided a study to proof this statement in line 57.

References:

[1] L. Xu, F. Wan, H. Fu, B. Tang, Z. Ruan, Y. Xiao, Q. Luo, Lactobacillus reuteri I5007 Modulates Intestinal Host Defense Peptide Expression in the Model of IPEC-J2 Cells and Neonatal Piglets, Microbiol Spectr, 2022, 10.

 [2] D.O. Morris, A. Loeffler, M.F. Davis, L. Guardabassi, J.S. Weese, Recommendations for approaches to meticillin‐resistant staphylococcal infections of small animals: diagnosis, therapeutic considerations, and preventative measures., Vet Dermatol, 2017, 28, 304.

  1. Line 77: Replace the word "repeats" with "replications".

Response: Thank you for your suggestions. We have modified this sentence in line 92.

  1. Line 83: Replace the word "basic" with "basal".

Response: Thank you for your comments. We have modified this sentence in line 98.

  1. Line 83: Replace the word "powder" with "mash".

Response: Thank you for your comments. We have modified this sentence 98.

  1. Lines 73 - 95. What is the experimental design?

Response: Thanks for your suggestion. We have modified this sentence 103. And the graphic illustration of the animal experiment design is shown in Supporting information Figure S1.

Reviewer 2 Report

Comments and Suggestions for Authors

This study found that dietary supplementation with lysozyme-cinnamaldehyde conjugates enhances feed conversion efficiency by improving intestinal health and modulating the gut microbiota in weaned piglets infected with enterotoxigenic Escherichia coli. The study trial was well designed with five treatment groups, including a negative control group, and an antibiotic positive control group. The study measured many indicators. That's a good study. My comments are as follows:

Title: "... enhances growth performance by improving...", but in Table 4, only one indicator, F/G, which is an indicator of feed conversion efficiency, was significantly different. But the average daily gain (ADG), a key growth performance indicator, was not significant (P > 0.05). So I suggest that the title can change to "Dietary supplementation with lysozyme-cinnamaldehyde conjugates enhances feed conversion efficiency by improving intestinal health and modulating the gut microbiota in weaned piglets infected with enterotoxigenic Escherichia coli".

Line 208: there are two periods at the end of this line. Please check the full text for formatting and grammatical errors, as well as punctuation.

Figure 7 and 8: The graph is light and the transparency needs to be adjusted.

Author Response

Dear Editor-in-chief,

Thank you very much for your and the reviewers’ effort on our manuscript (animals-2656122, entitled " Dietary supplementation with lysozyme-cinnamaldehyde conjugates enhances growth performance by improving intestinal health and modulating the gut microbiota in weaned piglets "). We have addressed the comments of the reviewer point-by-point and hope the revised version is now acceptable for publication in "Animals". The revised manuscript was marked with red color.

For your information, we have revised the manuscript accordingly as follows:

REVIEWER REPORT(S):

Referee: 2

Comments to the Author
This study found that dietary supplementation with lysozyme-cinnamaldehyde conjugates enhances feed conversion efficiency by improving intestinal health and modulating the gut microbiota in weaned piglets infected with enterotoxigenic Escherichia coli. The study trial was well designed with five treatment groups, including a negative control group, and an antibiotic positive control group. The study measured many indicators. That's a good study.

Major revision

1.Title: "... enhances growth performance by improving...", but in Table 4, only one indicator, F/G, which is an indicator of feed conversion efficiency, was significantly different. But the average daily gain (ADG), a key growth performance indicator, was not significant (P > 0.05). So, I suggest that the title can change to "Dietary supplementation with lysozyme-cinnamaldehyde conjugates enhances feed conversion efficiency by improving intestinal health and modulating the gut microbiota in weaned piglets infected with enterotoxigenic Escherichia coli".

Response: Thank you for our manuscript and for the constructive comments, with greatly helped us to improve the manuscript. We have taken the comments and we have modified the title to "Dietary supplementation with lysozyme-cinnamaldehyde conjugates enhances feed conversion efficiency by improving intestinal health and modulating the gut microbiota in weaned piglets infected with enterotoxigenic Escherichia coli".

  1. Line 208: there are two periods at the end of this line. Please check the full text for formatting and grammatical errors, as well as punctuation.

Response: Thank you for your suggestions. We have modified this sentence 208.

  1. Figure 7 and 8: The graph is light, and the transparency needs to be adjusted.

Response: Thank you for your suggestions. We have changed the color for Figure 7 and 8 to make them readable.